# The Expression Level of HIV-1 Vif Is Optimized by Nucleotide Changes in the Genomic SA1D2prox Region during the Viral Adaptation Process

**DOI:** 10.3390/v13102079

**Published:** 2021-10-15

**Authors:** Takaaki Koma, Naoya Doi, Mai Takemoto, Kyosuke Watanabe, Hideki Yamamoto, Satoshi Nakashima, Akio Adachi, Masako Nomaguchi

**Affiliations:** 1Department of Microbiology, Tokushima University Graduate School of Biomedical Sciences, Tokushima 770-8503, Japan; tkoma@tokushima-u.ac.jp (T.K.); naoya@tokushima-u.ac.jp (N.D.); maimaitakemoto@gmail.com (M.T.); watanabekyousuke8888@gmail.com (K.W.); johou.hideki.pokopoko@gmail.com (H.Y.); sterling.bertland@gmail.com (S.N.); 2Faculty of Medicine, Tokushima University, Tokushima 770-8503, Japan; 3Department of Microbiology, Kansai Medical University, Osaka 573-1010, Japan

**Keywords:** HIV-1, Vif expression, SA1D2prox, nucleotide sequence, adaptation, splicing

## Abstract

HIV-1 Vif plays an essential role in viral replication by antagonizing anti-viral cellular restriction factors, a family of APOBEC3 proteins. We have previously shown that naturally-occurring single-nucleotide mutations in the SA1D2prox region, which surrounds the splicing acceptor 1 and splicing donor 2 sites of the HIV-1 genome, dramatically alter the Vif expression level, resulting in variants with low or excessive Vif expression. In this study, we investigated how these HIV-1 variants with poor replication ability adapt and evolve under the pressure of APOBEC3 proteins. Adapted clones obtained through adaptation experiments exhibited an altered replication ability and Vif expression level compared to each parental clone. While various mutations were present throughout the viral genome, all replication-competent adapted clones with altered Vif expression levels were found to bear them within SA1D2prox, without exception. Indeed, the mutations identified within SA1D2prox were responsible for changes in the Vif expression levels and altered the splicing pattern. Moreover, for samples collected from HIV-1-infected patients, we showed that the nucleotide sequences of SA1D2prox can be chronologically changed and concomitantly affect the Vif expression levels. Taken together, these results demonstrated the importance of the SA1D2prox nucleotide sequence for modulating the Vif expression level during HIV-1 replication and adaptation.

## 1. Introduction

The HIV-1 Vif protein antagonizes host intrinsic restriction factors APOBEC3 (A3) proteins (A3DE, A3F, A3G, and A3H haplotype II/V/VII), which are cytidine deaminases [1,2,3,4,5,6,7]. Among the A3 proteins, A3G displays the strongest antiviral activity. In the absence of Vif, A3G proteins are incorporated into virions and inhibit HIV-1 replication mainly by introducing lethal G-to-A hypermutations into the viral genome in a deaminase-dependent manner [1,2,3,4,5,6,7,8]. Deaminase-independent restriction by A3G to the HIV-1 replication process, reverse transcription and integration, has been also demonstrated [4,5,6,7]. Vif works as an adaptor to recruit A3 proteins into the E3 ubiquitin ligase complex and leads to proteasome degradation to evade from their restriction [1,2,3,4,5,6,7,8]. Vif also inhibits transcription, translation, and packaging of A3G in a degradation-independent manner [8]. Once a small number of A3G molecules are incorporated into virions, A3G exerts potent antiviral effects [9,10]. Thus, the ability of Vif to counteract A3G is essential for HIV-1 replication especially in A3 proteins-expressing cells such as CD4+ T cells and macrophages.

HIV-1 mRNA species are produced through alternative splicing to express various viral proteins [11,12,13]. Although a variety of HIV-1 mRNA species are produced by the combination of splicing donors (SD1 to SD4) and splicing acceptors (SA1 to SA7), *vif* mRNA is generated by utilizing SD1 and SA1 sites (Figure 1A) [11,12,13,14]. Generally, splicing events are accomplished by an interplay between various host proteins and RNA secondary structures/*cis*-acting sequences in transcripts [15,16,17]. *Cis*-acting sequences, called splicing regulatory elements (SREs), are divided into two groups, enhancers and silencers, and recruit sequence-specific RNA binding factors [15,16,17]. SREs that are involved in *vif* transcript production and/or exon2 (the fragment from SA1 to SD2) inclusion have also been found within the upstream region of the *vif* start codon in the HIV-1 genome (Figure 1B) [18,19,20,21], (for review, see [16]). The HIV-1 alternative splicing process usually follows the cellular pre-mRNA splicing mechanism. However, unlike cellular mRNA production in which introns are completely removed, HIV-1 must maintain the full-length of RNA that is packaged into the progeny virion as the genome RNA and is transcribed into Gag/Gag-Pol proteins. Furthermore, well-balanced alternative splicing to ensure optimal HIV-1 mRNA production is also required for viral protein expression. The disruption of this process adversely affects HIV-1 replication. Thus, understanding the mechanism of the regulation of HIV-1 mRNA production is important for the control of viral replication [16,17].

HIV-1 has a high ability to adapt to various environments by acquiring mutations through the high mutation rate of reverse transcriptase (RT) and recombination of viral genomes [22,23]. The highly mutable nature of HIV-1 leads to the generation of various mutants that exhibit drug resistance, evasion from the host immune response, and transmission to new hosts [24,25,26,27]. In cases of drug-resistant mutants, they do not always acquire mutations in the sites targeted by drugs. For example, the development of resistance against a protease inhibitor (PI) has been reported to result from the emergence of mutations in the Gag or Envelope (Env) region but not in the Pol-protease region [28,29,30,31,32]. Even in Vif-null HIV-1 clones, adaptive clones that emerged during an adaptation experiment developed A3G resistance without Vif expression [33,34], although there is a report that a Vif-null HIV-1 clone failed to do so [35]. For a Vif-null HIV-1 adaptation, acquisition of mutations in Env has been shown to affect the Gag-Pol packaging and result in the incorporation of a high level of RT into virions, thereby enabling an escape from A3G restriction [33]. It has also been reported in another adaptation experiment that single-nucleotide mutations upstream of the Gag start codon increase virion production, resulting in dilution of the A3G-suppressive effects [34]. Adaptation experiments under restrictive conditions are an effective means to newly identify the site(s)/region(s) involved in evasion from the restriction and to understand a viral adaptation pathway and its mechanism.

Based on comparative sequence analyses of adaptive mutations that emerged in our HIV-1 adaptation system [36], we have identified a number of naturally-occurring single nucleotide mutations (nSNM) that vary the Vif expression levels within a region around the SA1 and SD2 sites located within the *pol*-integrase sequence, named the SA1D2prox region (Figure 1A). These nSNMs within SA1D2prox that alter the Vif expression levels also affect the replication potential depending on both their Vif expression levels and cellular A3G expression levels [37,38,39,40]. Pro-viral clones carrying nSNM are grouped into low-, high-, and excessive-Vif types, which express low, high, and extremely high levels of Vif, respectively. Of these, low- and excessive-Vif types exhibit a decreased growth ability compared to that of a wild-type (WT) clone in cells expressing a high level of A3G [38]. In this study, through adaptation experiments of the low- and excessive-Vif type HIV-1 clones under the A3G restrictive condition, we aimed to characterize the viral adaptation process, especially focusing on identifying novel site(s)/region(s) that affect the Vif expression level. To clarify the relationship between SA1D2prox and Vif expression during in vivo replication, we analyzed the effect of changes in the nucleotide sequence within SA1D2prox on the alteration in Vif expression levels using clinical samples obtained in chronological order. The findings in the present study highlight the importance of the SA1D2prox genomic sequence in determining the Vif expression levels.

## 2. Materials and Methods

### 2.1. Plasmids

Pro-viral clones (WT NL4-3, NL-ΔVif clone NL-Nd, NL-D229gat, NL-R224cgc, and NL-P238ccg) and a FLAG-tagged A3G expression vector have been described previously [38,41,42,43,44]. To construct the pcNLmini-RI clone, the following fragments were fused by overlapping PCR: fragment 1 (NL4-3 nucleotide positions 675 to 804 including the *Xba*I site at the 5′ end) and fragment 2 (NL4-3 nucleotide positions 4842 to 5748 including the authentic *Eco*RI site at the 3′ end), and then the resultant fragment was introduced into pcDNA3.1(-) vector (Invitrogen, Waltham, MA, USA). Mutant clones for NL4-3 and for pcNLmini-RI were generated by site-directed mutagenesis. 

### 2.2. Cells

A human embryonic kidney cell line, HEK 293T [45], was cultured in minimal essential medium supplemented with 10% heat-inactivated fetal bovine serum. A human lymphocyte cell line, H9, was cultured in RPMI 1640 supplemented with 10% heat-inactivated fetal bovine serum. 

### 2.3. Adaptation Experiments

The adaptation experiments performed are outlined in Figure 1. Virus stocks were prepared from HEK 293T cells transfected with pro-viral clones by the calcium phosphate coprecipitation method [41,46]. The virion-associated RT activity was measured as previously described [47,48]. Briefly, culture supernatant was mixed with an RT reaction mixture containing poly (A) as a template, oligo (dT)_18_ as a primer, and [α-^32^P]dTTP. After incubation, the reaction mixture was spotted onto a positively charged nylon membrane (Roche). The membrane was washed with 2 × SSC to remove unincorporated [α-^32^P]dTTP, and spots were then counted by scintillation counter. Equal units of RT activity (10^4^) were inoculated into H9 cells (10^6^) for the adaptation experiments of NL-D229gat, NL-R224cgc, and NL-P238ccg. The long-term culture of infected cells was carried similarly to that described previously [36]. Fresh H9 cells were added to the culture twice and three times for NL-D229gat and NL-R224cgc/NL-P238ccg, respectively, during the long-term culture in order to rescue adapted viruses. To construct adapted pro-viral clones, fragments from the *Sbf*I site to the *Bsa*BI site (Figure 1A) were amplified by PCR and introduced into the corresponding sites of the NL4-3 clone. For selected clones, sequence analyses were done for the region from the *Sbf*I to the *Bsa*BI sites. 

### 2.4. Infection Experiments

Virus stocks were prepared from HEK 293T cells transfected with pro-viral clones by the calcium phosphate coprecipitation method [41,46] or by Lipofectamine2000 (Invitrogen). The virion-associated RT activity was measured, and equal RT units were inoculated into H9 cells for the analysis of viral clones. Virus replication was monitored by the activity of RT released into the culture supernatants.

### 2.5. Western Blotting Analysis

Western blotting analysis was performed as described previously [37,38]. Briefly, pro-viral clones or pcNLmini-RI-based clones were transfected into HEK 293T cells by Lipofectamine2000, and unless otherwise noted, on day 1, post-transfection cell lysates were prepared with 1 × TNE buffer. Analyses of Vif and A3G expression levels in virions and cells were carried out similarly as described previously [43,44]. Briefly, HEK 293T cells were co-transfected with pro-viral clones along with a FLAG-tagged A3G expression vector by the calcium phosphate coprecipitation method. On day 1 post-transfection, virions were collected by ultracentrifugation, and the lysates of virions and cells were prepared with 1 × TNE buffer. Anti-HIV-1 Vif 319 (catalog no. ab66643; Abcam, Tokyo, Japan), anti-Gag-p24 (#3537, NIH Research and References Reagent Program), anti-FLAG (FLA-1, MEDICAL & BIOLOGICAL LABORATORIES), and anti-β-actin clone AC-15 (Sigma-Aldrich, Burlington, MA, USA) antibodies were used for immunoblotting analyses. 

### 2.6. Semiquantitative RT-PCR Analysis of Splicing Products

HEK 293T cells were transfected with pcNLmini-RI or its variant clones. As described previously [38,39], at 16 hrs post-transfection, cells were lysed for RNA extraction, and subsequent cDNA synthesis using oligo (dT) primer. PCR amplification was done similarly as described previously utilizing the following primer sets: XbaI-NL675-5 (Forward) (GCTCTAGAGAGGAGATCTCTCGACGCAG) and NL741-3 (Reverse) (GTCGCCGCCCCTCGCCTC) for all transcripts; XbaI-NL675-5 (Forward) and Vif-qPCR-3 (Reverse) (ACCTGCCATCTGTTTTCCATA) for the full-length and D1/A1 splicing products; XbaI-NL675-5 (Forward) and NL5622-3 (Reverse) (GCTCTAGTGTCCATTCATTG) for the D1/A1-D2/A2 and D1/A2 splicing products. PCR amplicons were resolved by Metaphor agarose gel (Lonza) and analyzed by the Amersham Imager 600 instrument (GE Healthcare, Chicago, IL, USA) [38].

## 3. Results

### 3.1. Viral Clones Used for the Adaptation Experiments

We have reported various nSNMs within the SA1D2prox region of the HIV-1 (NL4-3) genome that significantly affect the Vif expression level and viral replication potential (Figure 1A) [37,38,39,40]. Of these mutant clones, NL-D229gat and NL-R224cgc/NL-P238ccg are categorized into low-Vif and excessive-Vif types, respectively (Figure 1B). To evaluate Vif expression kinetics, HEK 293T cells were transfected with pro-viral clones NL4-3, NL-D229gat, NL-R224cgc or NL-P238ccg. On day 1 and day 2 post-transfection, the cells were lysed and subjected to Western blotting analysis. As shown in Figure 2A, Vif expression levels in NL-D229gat and NL-R224cgc/NL-P238ccg were very low and significantly high, respectively, compared to that in WT NL4-3. The difference in Vif expression levels among these clones was maintained at different time points of post-transfection. We thus decided to carry out experiments using samples prepared on day 1 post-transfection thereafter. To determine the effect of variation in Vif expression level on A3G antagonizing activity, pro-viral clones along with a FLAG-tagged A3G expression vector were co-transfected into HEK 293T cells. An NL-ΔVif clone was used for the negative control, and a WT NL4-3 clone was used for the positive control. On day 1 post-transfection, the lysates of virions and cells were prepared for Western blotting analysis (Figure 2B). For a negative control NL-ΔVif, significant amounts of A3G were observed in virions and cells relative to that for WT NL4-3. The A3G expression levels in virions and cells from low-Vif type NL-D229gat were similar to those in NL-ΔVif and significantly higher than those in WT NL4-3 and excessive-Vif type NL-R224cgc/NL-P238ccg (Figure 2B). A3G expression levels in virions and cells from excessive-Vif type NL-R224cgc/NL-P238ccg were similar to those in WT NL4-3, whereas Vif levels for the excessive clones were substantially higher than those for the other clones in cells and virions (Figure 2B). The results in Figure 2 indicate that low-Vif type NL-D229gat is not able to efficiently degrade A3G, and that, while excessive-Vif type NL-R224cgc/NL-P238ccg retain A3G antagonizing activity, large amounts of Vif were incorporated into virions of the two viral clones. A high level of HIV-1 Vif expression has been shown to inhibit viral infectivity [49] and to reduce virion production via the decrease in accumulation of *gag-pol* mRNA [19,50]. Indeed, we have shown that excessive-Vif type NL-R224cgc/NL-P238ccg exhibit reduced virus infectivity and virion production, and thus their viral replication abilities are decreased compared to WT NL4-3 [38]. Thus, the three variant clones grew poorly compared to the WT NL4-3 clone in the highly A3G-expressing H9 cells, which also express A3DE and A3F [38,51]. H9 cells have been reported to express A3G level similar to that in PBMC, though individual differences were observed [38,52,53]. In this study, we performed adaptation experiments using the low- and excessive-Vif type viral clones and the potently virus-restrictive H9 cell line as host cells (Figure 1C). To construct adapted pro-viral clones, we utilized the fragment from the *Sbf*I site to the *Bsa*BI site of the obtained adapted viruses to maintain a series of SD and SA sites, except for the SD1 and SA7 sites, on the viral genome (Figure 1A). Through analyses of the adapted clones, we specifically explored the sites involved in the alteration of the Vif expression level. 

### 3.2. Characteristics of Viral Clones Obtained from Low-Vif Type NL-D229gat Adaptation

We performed a long-term culture of NL-D229gat-infected H9 cells as outlined in Figure 1C, and constructed adapted (NL-gatad) clones as described above. To test the replication-competence of NL-gatad clones, a multi-cycle infection experiment was performed (Figure 3A). Viruses prepared from HEK 293T cells transfected with a WT NL4-3, a parental NL-D229gat, or seven NL-gatad clones were inoculated into H9 cells. Virus replication was monitored by RT activity in the culture supernatants collected every 3 days post-infection. As shown in Figure 3A, consistent with our previous result [38], the NL-D229gat displayed significantly delayed kinetics relative to that of NL4-3. All the NL-gatad clones tested, except for NL-gatad-5, exhibited higher growth potentials than the parental NL-D229gat, albeit still lower relative to NL4-3. To examine the alteration in the Vif expression levels, these clones were transfected into HEK 293T cells. As shown in Figure 3B, the Vif expression levels of NL-gatad clones 1, 5, 6, and 7 were clearly increased compared to that of NL-D229gat, whereas NL-gatad clones 4, 9, and 10 displayed similar expression levels to that of NL-D229gat. We carried out sequence analysis of several NL-gatad clones (1, 4, 6, 7, and 9). NL-gatad clones 4 and 9, whose Vif expression levels were not altered, carry several mutations in the Vif, Vpu/Env, and Env regions (Table 1). As for NL-gatad-9, while a mutation (D270gac) was found within the SA1D2prox region, we have shown previously that the D270gac mutation does not significantly affect the *vif* transcript level (0.96 on average relative to NL4-3) [39]. Thus, clones 4 and 9 exhibited enhanced growth potential probably by acquiring mutation(s) without an increase in the Vif expression levels. On the other hand, various mutations are present throughout the genomes of NL-gatad clones 1, 6, and 7, which exhibited the increased Vif expression level (Table 2). Most importantly, all three clones carried the R269Kaag mutation within the SA1D2prox region. This mutation, a natural variation found in the G_I2-1_ element, has been experimentally shown to increase the Vif expression level (Figure 1B) [38].

### 3.3. Characteristics of Viral Clones Obtained from Excessive-Vif Type NL-R224cgc Adaptation

We constructed adapted clones (NL-cgcad) derived from a long-term culture of NL-R224cgc-infected cells as described above (Figure 1C), and examined their replication abilities along with WT NL4-3 and the parental NL-R224cgc clones (Figure 4A). As we have shown previously [38], NL-R224cgc displayed moderately reduced growth potential relative to NL4-3. The replication ability of three NL-cgcad clones 3, 7, and 10 was comparable to that of WT NL4-3, and NL-cgcad-1 grew better than NL-R224cgc, but to a lesser extent compared to NL4-3 (note the peak days for RT production). As for the growth kinetics of the other clones, NL-cgcad-4 and NL-R224cgc grew similarly, whereas clones 2, 6, and 8 displayed poor growth compared to NL-R224cgc (Figure 4A). Of the eight adapted clones tested, the Vif expression level of clones 1, 2, and 4 was markedly reduced to a varying degree relative to those for both NL-R224cgc and NL4-3 (Figure 4B). The sequences of NL-cgcad clones 3, 7, and 10 are identical to that of NL4-3, suggesting a reversion of R224cgc to R224cgg (Figure 1B). NL-cgcad clones 1, 2, and 4, which displayed a remarkably reduced level of Vif expression, carry some mutations (Y227tat, D229gat, and I267att) within the SA1D2prox region (Table 3). Of these mutations, D229gat is the same mutation used in this study (Figure 1 and Figure 3), and the I267att mutation moderately reduced the *vif* transcript level (0.71 on average relative to NL4-3) [39]. NL-cgcad clones 1, 2, and 4 have many mutations including a Vif deletion, a Vpr deletion, and Env mutations (Table 3). The difference in growth potential among NL-cgcad clones 1, 2, and 4 may be due to the Vif expression level and/or other mutation(s), which can affect the viral replication ability. 

### 3.4. Characteristics of Viral Clones Obtained from Excessive-Vif Type NL-P238ccg Adaptation

We examined the replication potential of adapted clones (NL-ccgad) obtained through an adaptation experiment of NL-P238ccg (Figure 1C). As shown in Figure 5A, NL-P238ccg grew poorly compared to NL4-3, as we described previously [38]. The growth kinetics of NL-ccgad clones 5, 8, 10, and 13 were almost the same as that of NL4-3. While an enhanced growth potential of NL-ccgad-6 was observed compared to that of a parental NL-P238ccg, NL-ccgad clones 1 and 7 exhibited a reduced replication ability (Figure 5A). The Vif expression levels for most of the NL-ccgad clones tested were similar to that of WT NL4-3, whereas NL-ccgad clones 1 and 6 showed a considerably reduced level of Vif expression (Figure 5B). We carried out sequence analyses of NL-ccgad clones 5, 8, and 10 which grow as well as NL4-3. These three clones are revertants that have P238cca instead of P238ccg, and also carry several synonymous and non-synonymous mutations within the Vif, Vpr, and Env regions (data not shown). As shown in Table 4, low-Vif type NL-ccgad clones 1 and 6 carry mutations, G237ggg and D229gat, respectively, within the SA1D2prox region. As mentioned above, the Vif expression level was decreased by the D229gat mutation (Figure 3B). The difference in growth potential between NL-ccgad clones 1 and 6 may be explained by the presence of mutation(s) that can affect the Vif expression level and/or viral replication ability.

### 3.5. Identification of Adaptive Mutations Associated with the Variation in Vif Expression Level

In total, all adapted clones with the altered Vif expression level carry a mutation or mutations within the SA1D2prox region as follows: NL-gatad clones 1, 6, and 7 have the R269Kaag mutation; NL-cgcad clones 1, 2, and 4 carry either the Y227tat mutation or the D229gat/I267att double mutation; NL-ccgad clones 1 and 6 possess the G237ggg mutation and the D229gat mutation, respectively (Table 2, Table 3 and Table 4). To determine whether these mutations affect the Vif expression level, we newly constructed pro-viral clones carrying two mutations, a parental mutation and a mutation found in the SA1D2prox region. The Vif expression levels were analyzed using HEK 293T cells transfected with pro-viral clones (Figure 6). A pro-viral clone (NL-gat+aag), bearing D229gat (parental) and R269Kaag (adaptive) mutations, exhibited an increase in the Vif expression level compared to NL-D229gat, but to a lesser extent than NL4-3. The introduction of either Y227tat or D229gat into a parental NL-R224cgc clone (NL-cgc+tat and NL-cgc+gat) markedly reduced the Vif expression level relative to NL-R224cgc and NL4-3. Similarly, the Vif expression levels were remarkably decreased for clones NL-ccg+ggg and NL-ccg+gat carrying a parental P238ccg mutation and either G237ggg or D229gat (Figure 6). While we have already shown the effect of mutations R269Kaag and D229gat on the Vif expression level [38], we newly identified mutations Y227tat and G237ggg that strongly reduce the Vif expression level in this study. In this regard, we have previously shown that the *vif* mRNA expression level of the Y227Fttc mutant clone is similar to that of WT NL4-3 [38]. The single nucleotide change of WT Y227tac to either Y227tat or Y227Fttc differently affects the Vif expression level. This result implies the importance of nucleotide sequence in the SA1D2prox region, although whether the Y227tat mutation alone can alter the Vif expression level needs to be examined. In sum, the change in the Vif expression levels of five newly constructed pro-viral clones (Figure 6) were well correlated with the alteration in those of adapted viral clones (Figure 3, Figure 4 and Figure 5). These results indicate that mutations found within the SA1D2prox region are indeed adaptive mutations, suggesting a strong association between the nucleotide sequence of the SA1D2prox region and the determination of the Vif expression level.

### 3.6. Effect of Adaptive Mutations within the SA1D2prox Region on the Splicing Pattern

We have previously demonstrated that nSNMs within the SA1D2prox region do not affect the total HIV-1 mRNAs production levels but change the overall splicing pattern, which in turn alters the *vif* mRNA/Vif protein expression level and viral growth potential [37,38,39]. Moreover, we and others noticed the inverse correlation between *vif* mRNA/protein and *vpr* mRNA/protein expression levels [37,38,54]. To facilitate evaluation of the changes in the splicing pattern, we newly constructed a minigenome (pcNLmini-RI) possessing SA1 and SA2 that are important for *vif* and *vpr* mRNA production, respectively (Figure 7A). The minigenome contains the SD1 site, the SA1D2prox region, the entire *vif* sequence, and the partial *vpr* sequence from the Vpr start codon to the authentic *Eco*RI site of NL4-3. Combinatorial mutation sets of parental mutations (D229gat, R224cgc, and P238ccg) and adaptive mutations (R269Kaag, Y227tat, D229gat, and G237ggg) identical to those in the pro-viral clones described above were introduced into the minigenome to analyze the splicing pattern. We first ascertained whether the minigenomes constructed display the altered Vif expression levels as observed for pro-viral clones (Figure 6). The Vif expression level for minigenomes was determined using transfected HEK 293T cells (Figure 7B). In agreement with the results obtained using pro-viral clones (Figure 6), the Vif expression levels varied among minigenomes carrying the mutations (Figure 7B). A minigenome carrying both D229gat and R269Kaag exhibited an increased level of Vif expression relative to that with D229gat alone. The reduction in Vif expression levels was evident for minigenomes containing an adaptive mutation of Y227tat, D229gat, or G237ggg relative to the parental R224cgc and P238ccg constructs. The alteration in the Vif expression level for the minigenomes tested correlated very well with that of pro-viral clones (Figure 6 and Figure 7). 

Next, we determined the change in splicing patterns by mutations using the minigenome constructs. HEK 293T cells were transfected with various minigenomes, cells were lysed for RNA extraction, and then cDNA samples were synthesized using oligo(dT) primer. Semiquantitative PCR was carried out to amplify distinct 200–400 bp splicing products (full-length, D1/A1, D1/A1-D2/A2, and D1/A2) using the specific primer sets depicted in Figure 8A. D1 (Figure 8A) and the *gapdh* products were also amplified simultaneously, and analyzed as controls for transfection efficiency and total RNA amount, respectively. As shown in Figure 8B, the low-Vif type D229gat displayed a decrease in the D1/A1 splicing product and an undetectable level of D1/A1-D2/A2 product relative to WT without any mutations, whereas the expression levels of full-length and D1/A2 products in D229gat were increased (Figure 8B). As expected, the D1/A1 product was increased in a minigenome with both D229gat and R269Kaag (gat+aag) compared to a parental minigenome D229gat. While the D1/A1-D2/A2 product for the gat+aag construct was still undetectable, a decrease in full-length and D1/A2 products was observed relative to D229gat (Figure 8B). For excessive-Vif type R224cgc and P238ccg, the changes in the splicing pattern by adaptive mutations were also observed depending on the alteration in the Vif expression level (Figure 8C). The D1/A1 and D1/A1-D2/A2 products by R224cgc and P238ccg were elevated compared to that of WT, whereas a decrease in full-length and D1/A2 products was observed for both of them. Minigenomes carrying an adaptive mutation along with a parental mutation (R224cgc+Y227tat, R224cgc+D229gat, P238ccg+D229gat, and P238ccg+G237ggg) exhibited decreased levels of D1/A1 and D1/A1-D2/A2 products but increased levels of full-length and D1/A2 products relative to each parental minigenome (Figure 8C). These results, in good agreement with previous reports, showed that while the change in D1/A1 production level by mutations in the SA1D2prox correlates with the alteration in the Vif expression level, the D1/A2 production level is inversely correlated with the D1/A1 production level. Taken together, these results suggest that adaptive mutations identified within the SA1D2prox region which alter the Vif expression level affect the mRNA production levels mainly via effects on the splicing.

### 3.7. Chronological Changes in the Vif Expression Levels by Mutations within SA1D2prox in Infected Individuals

We have previously found various nSNMs that can alter the Vif expression level by examining the SA1D2prox sequence obtained from the HIV-1 sequence database (Los Alamos National Laboratory) [37,38,39]. It was of interest, in the present study, to determine whether changes in the nucleotide sequence of SA1D2prox can be observed for samples obtained chronologically from patients and whether the changes influence the Vif expression level. For this purpose, we used the next-generation sequence data from plasma samples collected from protease inhibitor (PI)-resistant and integrase inhibitor (RAL)-resistant patients at different time points [55]. Pro-viral clones designated NL-pC2 and NL-pC3, which carry two and three nucleotide variations relative to NL4-3, respectively, represent a consensus sequence of SA1D2prox in HIV-1 subtype B strain (Figure 9A) [39]. The *vif* transcript levels for NL-pC2 and NL-pC3 were similar and slightly lower than that for NL4-3 (around 0.7 for NL-pC3 relative to that of NL4-3) [39]. The sequence data of SA1D2prox were obtained at 10 and 6 different time points for PI-resistant patients PI1and PI4, respectively (Figure 9B,C), and at 7 and 3 different time points for RAL-resistant patients RAL4 and RAL5, respectively (Figure 9D,E). Changes in the SA1D2prox sequence were observed over the survey period for the PI- and RAL-resistant patient samples used. Pro-viral clones that have the nucleotide sequence of SA1D2prox obtained were constructed using NL4-3 as a backbone, and were divided into groups (A–C or D) based on the sequence identity, e.g., PI1-A and PI1-B (Figure 9B–D). These pro-viral clones were transfected into HEK 293T cells, and cell lysates prepared were subjected to Western blotting analyses. As shown in Figure 10, although no significant alteration was observed for the Vif expression levels among PI1-A to PI1-C and PI4-A to PI4-D samples, these clones exhibited a moderately decreased level of Vif expression compared to their reference clones, NL-pC2 and NL-pC3. Importantly, these decreases in the Vif expression levels were expected, because we have already shown that many variations, if not all, found in the sequences for PI1 and PI4 samples lower the *vif* transcript/mRNA levels relative to NL4-3 [37,38]: D232gac (0.47), L242ctt (0.28), Q252cag (0.69), and K258aag (0.93) for PI1; R228aga (0.28), D253gac (1.03), R263agg (0.60), and K266aaa (0.75) for PI4. Interestingly, the Vif expression levels were different among the RAL4-A to RAL4-D and RAL5-A to RAL5-C samples, and varied during the survey period (Figure 10). For the RAL4 samples, the Vif expression level of RAL4-C was increased relative to that of RAL4-B. It is reasonably assumed that this may be due to the presence of V249gtg within the RAL4-C sequence which we have shown to increase the *vif* transcript level compared to NL4-3 (1.83 on average) [39]. These results showed that the SA1D2prox sequence can be chronologically changed and affect the Vif expression level during in vivo replication.

## 4. Discussion

We have previously reported that the low-Vif type NL-D229gat and excessive Vif-type NL-R224cgc/P238ccg clones exhibit a reduced growth potential relative to WT NL4-3 depending on their Vif expression levels and cellular A3G expression level (Figure 2) [37,38,39]. In this study, we sought to determine how these clones adapt under the growth restrictive condition imposed by H9 cells with a high expression level of A3G, especially focusing on the fluctuation in the Vif expression level. Some of the adapted clones constructed exhibited altered Vif expression levels relative to the parental clone (Figure 3, Figure 4 and Figure 5) and carry mutations within SA1D2prox (Table 1, Table 2, Table 3 and Table 4). Indeed, these mutations found in the region were responsible for the alteration in the Vif expression level (Figure 6). Minigenomes newly constructed recapitulated changes in the Vif expression level by parental and adaptive mutations observed for pro-viral clones. Semiquantitative PCR analyses were carried out using minigenomes to examine the effect of mutations on the splicing patterns. In agreement with previous reports [38,39,54], the alteration in the Vif expression level correlated well with the change in the D1/A1 splicing production levels, and there was an inverse correlation between the D1/A1 and D1/A2 production levels (Figure 7 and Figure 8). Moreover, we analyzed the Vif expression level using NL-based pro-viral clones that carry the SA1D2prox sequence obtained from plasma samples collected from patients at different time points. The results showed that, even in intra-patient samples, the SA1D2prox sequence can fluctuate chronologically and variations in the sequence can affect the Vif expression levels (Figure 9 and Figure 10). Taken together, the SA1D2prox nucleotide sequence is associated with viral adaptation through tuning the Vif expression level vital for the HIV-1 replication property.

We have previously proposed that there may be an appropriate Vif expression level required for efficient HIV-1 replication, because a viral clone with a low level of Vif expression fails to sufficiently degrade A3G, and because viral infectivity and virion production are inhibited by a high level of Vif expression [19,38,49,50]. Even though moderate alterations in *vif* production level by nSNMs within SA1D2prox (around 0.2~around 1.8 relative to WT NL4-3) do not significantly affect HIV-1 replication, low-Vif type NL-D229gat and excessive-Vif type NL-R224cgc/NL-P238ccg, which exhibit drastic changes in Vif level below 0.1 and over 2.5 for *vif* production relative to WT NL4-3, respectively, have the impaired replication ability as reported [38]. Adapted clones were generated from viruses that emerged in long-term cultures of H9 cells infected with low- and excessive-Vif type clones. Importantly, all replication-competent adapted viruses with altered Vif expression levels invariably harbored single-nucleotide changes within SA1D2prox. This implies that, during adaptation, HIV-1 quickly repairs an inappropriate range of Vif expression level, which resulted in an inferior viral replication, through acquiring single-nucleotide mutations, mostly synonymous mutations, within SA1D2prox. It has been reported that the introduction of numerous synonymous mutations throughout the HIV-1 genome, especially into the central region of the genome, results in replication defects via splicing perturbation [56]. Several lines of evidence showed that synonymous mutations can affect pivotal biological processes such as RNA splicing, RNA stability, and translation efficiency [56,57,58], for review, see [59,60]. This implies that synonymous mutations would be biologically relevant and exposed to selective forces. Thus, synonymous mutations can contribute to HIV-1 adaptation and evolution [60]. The results in this study provide a good example of this phenomenon. The SA1D2prox sequence contains splicing sites SA1/SD2 and is located within the Pol-integrase region, and thus amino acid changes by substitutions in the SA1D2prox sequence can affect the integrase function. For HIV-1 adaptation, it is reasonable to repair replication defects involved in Vif expression levels through synonymous mutations within the SA1D2prox region without adverse effects on the integrase function. 

The difference in growth potential was observed among adapted clones that exhibit similar Vif expression levels (Figure 3, Figure 4 and Figure 5). We found various mutations throughout the genomes of these adapted clones (Table 1, Table 2, Table 3 and Table 4). It is obvious that, since Vif expression level is not a sole determinant for HIV-1 replication ability in cells, some of these mutations can positively or negatively affect the viral replication ability independent of the Vif expression level. The effect of such mutations on the viral replication ability is currently under investigation in our laboratory. Another intriguing project is to investigate the effect of variation in cellular A3G expression level on an adaptation of HIV-1 clones with different Vif expression levels. This is ongoing research in our laboratory, in which we have been working on adaptation experiments using our SA1D2prox mutant clones and CEM-SS cells expressing a low level of A3G. 

Chronological alterations in Vif expression level by changes of SA1D2prox nucleotide sequence were confirmed also by using sequences of the region from patients’ samples. This result may imply that the SA1D2prox nucleotide sequence continues to change to retain an appropriate Vif level in individuals. It would be intriguing to study whether the increase and decrease in the Vif expression level by changing the SA1D2prox nucleotide sequence is associated with viral adaptation and replication ability in individuals. However, it may be difficult to investigate the involvement of the Vif expression level in viral replication ability/adaptation in vivo, considering that correlations between the expression level of A3G or A3F in individuals and the viral replication level/the disease progression state remain quite controversial [5,61]. In addition, changes in D1/A1 splicing can affect D1/A2 splicing, suggesting that the Vif expression level can be influenced by the nucleotide sequence of region(s) distinct from SA1D2prox. It would be harder to prove a role of the SA1D2prox sequence for in vivo replication via the effect of the Vif expression level.

The mechanism for HIV-1 mRNA splicing appears to be more complicated than previously thought, especially for *vif* and *vpr* mRNA production [17]. Going forward, further studies are required to elucidate the mechanism by which nSNMs found within SA1D2prox alter the *vif* mRNA/Vif protein expression levels. As we have shown, nSNMs within SA1D2prox alter the D1/A1 splicing product levels and concomitantly inversely change the D1/A2 splicing product levels (this study and [37,38,39]). In fact, we have confirmed the fluctuation in the Vif expression level for pro-viral clones that carry various chimeric *vif* sequences containing SA2 and SD3 between subtype B (NL4-3) and subtype C (IndieC) [40]. Given the long-range interaction between splicing sites [62], nucleotide sequences that affect the *vif* mRNA/Vif protein expression level would be present in various areas on the HIV-1 genome. Even if so, the results using the SA1D2prox sequence from patients suggested the possibility that the Vif expression level can be estimated based on the data for changes in the *vif* mRNA/Vif protein expression levels by individual mutations (Figure 9 and Figure 10). Once we can comprehensively identify mutations that affect the Vif expression levels, we may be able to predict the expression level just by looking at the sequence. Collectively, clarifying the significance of individual mutations experimentally identified is crucial for understanding viral replication, adaptation, and pathogenesis.

## Figures and Tables

**Figure 1 viruses-13-02079-f001:**
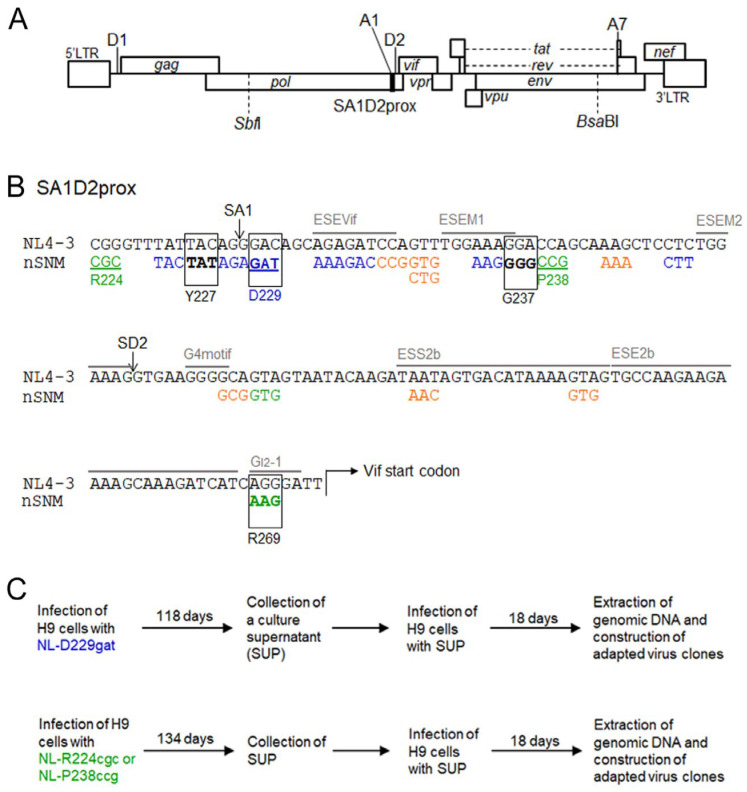
SA1D2prox sequence on the HIV-1 genome and virus adaptation procedure. (**A**) HIV-1 (NL4-3) genome organization. The SA1D2prox region is shown by a black box. Some of the splicing sites (SD1, SA1, SD2, and SA7) of the HIV-1 genome are indicated. Unique restriction enzyme sites (*Sbf*I and *Bsa*BI) used to make adapted viral clones are indicated. (**B**) SA1D2prox sequence. Sequences of low (NL-D229gat) and excessive (NL-R224cgc and NL-P238ccg) Vif types used for the adaptation experiments in this study are shown by blue and green letters/lines, respectively. Blue, orange, and green letters without underlines indicate single-nucleotide mutants identified by us [37,38,39] which display low, high, and excessive Vif types, respectively. Boxed, bold letters show adaptive mutations that change the Vif expression levels identified in this study. Splicing sites (SA1 and SD2) are indicated. Several known splicing enhancer/silencer motifs are represented by gray letters and lines [18,19,20,21], (for review, see [16]). nSNM, naturally-occurring single nucleotide mutation. (**C**) Virus adaptation. Viral clones of low (NL-D229gat) and excessive (NL-R224cgc and NL-P238ccg) Vif types were inoculated into H9 cells and cultured for the indicated periods. Fresh H9 cells were infected with the SUP collected on day 118 for NL-D229gat and day 134 for NLR224cgc/NL-P238ccg. On day 18 post-infection, cellular genomic DNA was extracted and used for the construction of adapted viral clones.

**Figure 2 viruses-13-02079-f002:**
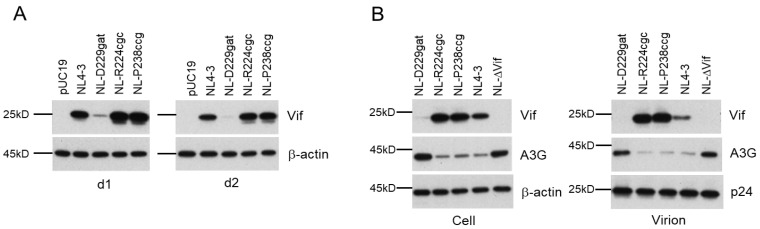
Characteristics of low-Vif type NL-D229gat and excessive-Vif type NL-R224cgc/NL-P238ccg for Vif expression level and A3G degrading activity. (**A**) Vif expression kinetics. HEK 293T cells were transfected with the indicated pro-viral clones. On day 1 and day 2 (d1 and d2, respectively) post-transfection, cell lysates were prepared and subjected to Western blotting analysis using anti-Vif and anti-β-actin antibodies. Representative data from two independent experiments are shown. (**B**) A3G antagonizing activity. The indicated pro-viral clones along with a FLAG-tagged A3G expression vector were co-transfected into HEK 293T cells. On day 1 post-transfection, virions and cells were collected and lysed for Western blotting analysis using anti-Vif, anti-Gag-p24, anti-FLAG, and anti-β-actin antibodies. Representative data from two independent experiments are shown.

**Figure 3 viruses-13-02079-f003:**
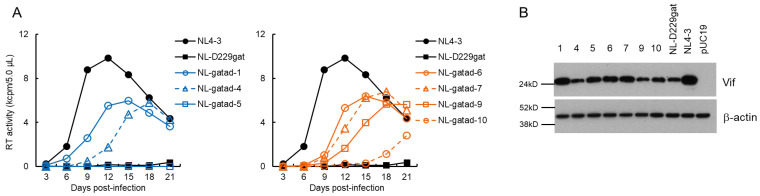
Characteristics of viral clones constructed from adapted NL-D229gat. (**A**) Growth kinetics. Viruses prepared from HEK 293T cells transfected with the indicated pro-viral clones were inoculated into H9 cells (1 × 10^4^ RT units/10^5^ cells). Virus replication was monitored by the virion-associated RT activity in the culture supernatants. This experiment was performed once to select viral clones that have the ability to grow in cells. Viral replication kinetics of WT NL4-3 and a parental NL-D229gat are presented in both panels for easy comparison. (**B**) Vif expression levels. HEK 293T cells were transfected with the indicated pro-viral clones. On day 1 post-transfection, cell lysates were prepared and subjected to Western blotting analysis using anti-Vif and anti-β-actin antibodies. An empty vector pUC19 and an authentic HIV-1 NL4-3 clone were used as negative and positive controls, respectively. Numbers correspond to those for the adapted clones shown in panel (**A**). Representative data from two independent experiments are shown.

**Figure 4 viruses-13-02079-f004:**
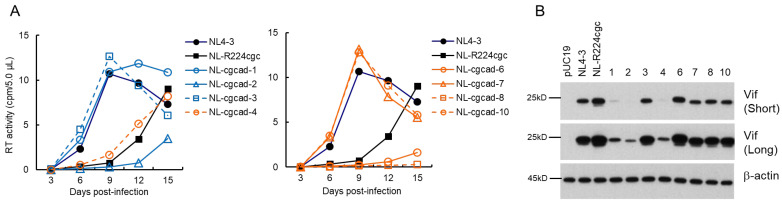
Characteristics of viral clones constructed from adapted NL-R224cgc. (**A**) Growth kinetics. Viruses prepared from HEK 293T cells transfected with the indicated pro-viral clones were inoculated into H9 cells (2 × 10^3^ RT units/10^5^ cells). Virus replication was monitored by the virion-associated RT activity in the culture supernatants. This experiment was performed once to select viral clones that have the ability to grow in cells. Viral replication kinetics of WT NL4-3 and a parental NL-R224cgc are presented in both panels for easy comparison. (**B**) Vif expression levels. Western blotting analysis was carried out as described in Figure 3B. Numbers correspond to those for the adapted clones shown in panel (**A**). Representative data from two independent experiments are shown. Short, short exposure; Long, long exposure.

**Figure 5 viruses-13-02079-f005:**
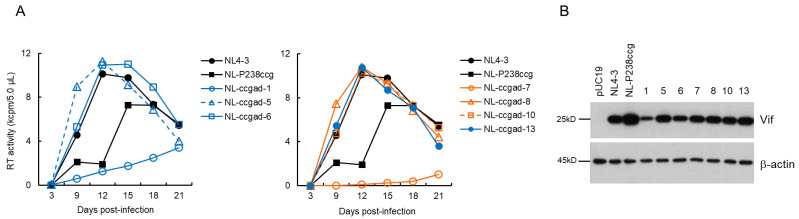
Characteristics of viral clones constructed from adapted NL-P238ccg. (**A**) Growth kinetics. Viruses prepared from HEK 293T cells transfected with the indicated pro-viral clones were inoculated into H9 cells (1 × 10^4^ RT units/10^5^ cells). Virus replication was monitored by the virion-associated RT activity in the culture supernatants. This experiment was performed once to select viral clones that have the ability to grow in cells. Viral replication kinetics of WT NL4-3 and a parental NL-P238ccg are presented in both panels for easy comparison. (**B**) Vif expression levels. Western blotting analysis was carried out as described in Figure 3B. Numbers correspond to those for the adapted clones shown in panel (**A**). Representative data from two independent experiments are shown.

**Figure 6 viruses-13-02079-f006:**
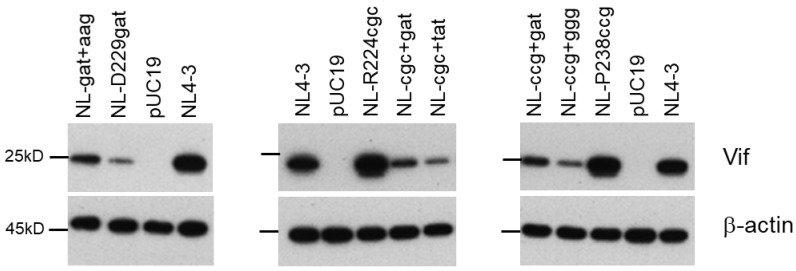
Effects of adaptive mutations within SA1D2prox on the Vif expression level. HEK 293T cells were transfected with the indicated pro-viral clones. On day 1 post-transfection, cell lysates were prepared for Western blotting analysis using anti-Vif and anti-β-actin antibodies. Adaptive mutations that change the Vif expression levels found within the SA1D2prox region of NL-D229gat, NL-R224cgc, and NL-P238ccg are shown in Table 2, Table 3 and Table 4, respectively. Representative data from two independent experiments are shown. gat, D229gat; aag, R269Kaag; cgc, R224cgc; tat, Y227tat; ccg, P238ccg; ggg, G237ggg.

**Figure 7 viruses-13-02079-f007:**
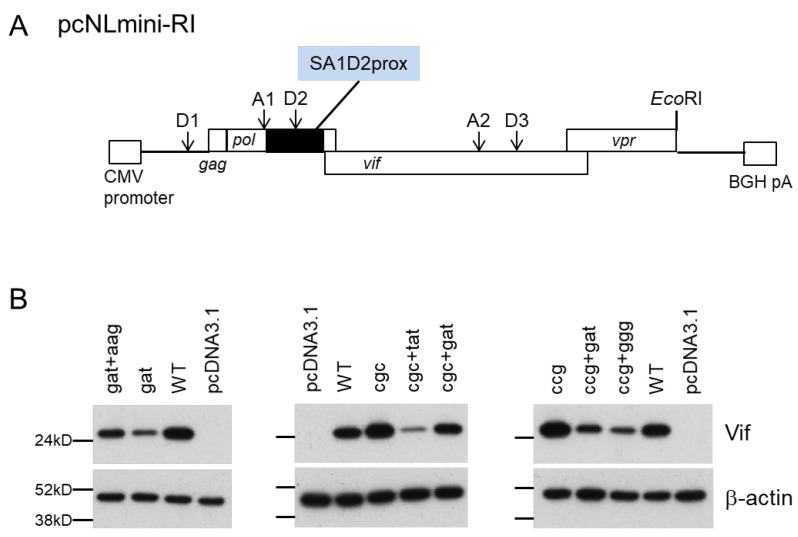
Vif expression levels of a newly constructed minigenome (pcNLmini-RI) vector. (**A**) Organization of the minigenome. The pcNLmini-RI vector was constructed as described in the Materials and Methods. Splicing (D1, A1, D2, A2, D3) and *Eco*RI sites in the minigenome vector corresponding to those in the authentic HIV-1 NL4-3 clone are shown. A black box indicates the SA1D2prox region. CMV promoter, cytomegalovirus enhancer-promoter; BGH pA, bovine growth hormone polyadenylation sequence. (**B**) Vif expression levels of the minigenome vectors carrying various mutations. HEK 293T cells were transfected with the indicated minigenome vectors, and on day 1 post-transfection, cell lysates were prepared for Western blotting analysis using anti-Vif and anti-β-actin antibodies. Mutations introduced into the minigenome vector were the same as those shown in Figure 5. An empty vector (pcDNA3.1) was used as a negative control. Representative data from two independent experiments are shown. WT, wild type; gat, D229gat; aag, R269Kaag; cgc, R224cgc; tat, Y227tat; ccg, P238ccg; ggg, G237ggg.

**Figure 8 viruses-13-02079-f008:**
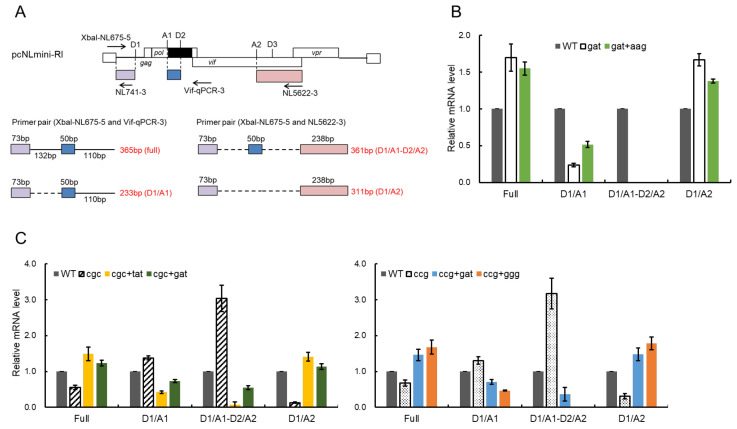
Effects of adaptive mutations on mRNA production. (**A**) mRNA produced from the minigenome. Splicing sites on the minigenome are presented. The primers used for the splicing pattern analysis are indicated by arrows with their names (upper part). Colored boxes show exons produced by splicing at various sites. In the lower part, splicing products analyzed using the primer sets are presented along with their lengths. Broken and solid lines show regions with and without splicing, respectively. (**B**,**C**) Changes in the splicing pattern of parental clones (gat, cgc, and ccg) by their adaptive mutations. Total RNA was prepared from HEK 293T cells transfected with the indicated minigenome vectors and subjected to semiquantitative RT-PCR using the primer pairs shown in (**A**). The signal intensities of semiquantitative RT-PCR products were quantitated from three independent experiments. The intensities of the indicated mRNAs in each sample were normalized to those of all viral mRNAs (D1) and *gapdh* mRNA. The normalized mRNA intensities in each sample relative to those of WT are presented.

**Figure 9 viruses-13-02079-f009:**
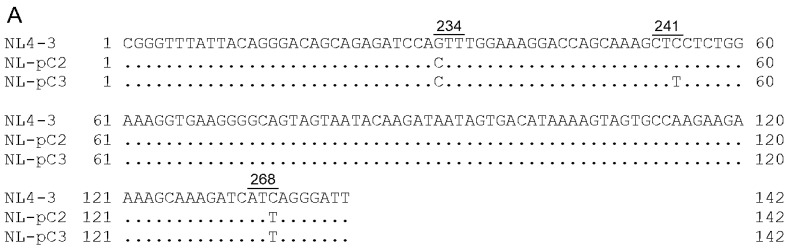
SA1D2prox sequences of NL4-3, its derivatives, and clinical samples. Sequence alignments were carried out using Genetyx Ver. 15. Numbers and underlines above the sequences represent Pol-IN amino acid positions (NL4-3 numbering) and codons, respectively. (**A**) Sequences of NL4-3, NL-pC2, and NL-pC3. NL-pC2 and NL-pC3 clones have 2 and 3 nucleotide substitutions, respectively, which are major nucleotides at each position in the SA1D2prox sequence within the HIV-1 subtype B population. (**B**,**C**) Chronological changes of the SA1D2prox sequence in samples obtained at different time points from PI-resistant patients PI1 (**B**) and PI4 (**C**). Sequence data were published previously [55] (DDBJ; PRJDB3502). Clone name was designated PI1-A, -B, and -C (**B**) and PI4-A, -B, -C, and -D (**C**) based on the sequence identity. (**D**,**E**) Chronological changes of the SA1D2prox sequence in samples obtained at different time points from RAL-resistant patients RAL4 (**D**) and RAL5 (**E**). Sequence data were published previously [55] (DDBJ; PRJDB3502). Clone name was designated RAL4-A, -B, and -C (**D**) and RAL5-A, -B, and -C (**E**) based on the sequence identity and the chronological order.

**Figure 10 viruses-13-02079-f010:**
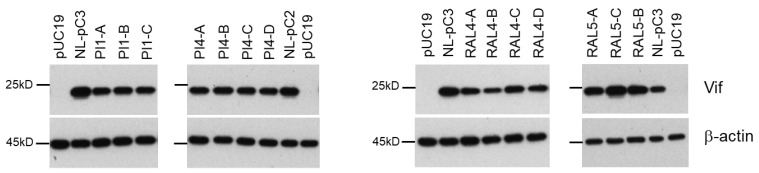
Vif expression levels of pro-viral clones carrying different SA1D2prox sequences in the HIV-1 NL4-3 genome. HEK 293T cells were transfected with the indicated pro-viral clones, and on day 1 post-transfection, cell lysates were prepared and subjected to Western blotting analysis using anti-Vif and anti-β-actin antibodies. Representative data from two independent experiments are shown.

**Table 1 viruses-13-02079-t001:** Mutations found in adapted clones which exhibit similar Vif expression levels to that in the NL-D229gat clone.

NL-gatad-4			NL-gatad-9		
Nt Change	Region	NS/S Change in the Region	Nt Change	Region	NS/N Change in the Region
c4916t	Pol-IN (SA1D2prox)	D229gat	c4916t	Pol-IN (SA1D2prox)	D229gat
			t5039c	Pol-IN (SA1D2prox)	D270gac
g5389a	Vif	E117K	g5389a	Vif	E117K
g6280a	Vpu	A74T	g6280a	Vpu	A74T
	Env	M20I		Env	M20I
g7016a	Env	E266K			
g7032a	Env	R271K			

Nt, Nucleotide; NS, non-synonymous; S, synonymous.

**Table 2 viruses-13-02079-t002:** Mutations found in adapted clones that exhibit higher Vif expression levels than that in the NL-D229gat clone.

NL-Gatad-1			NL-Gatad-6			NL-Gatad-7		
Nt Change	Region	NS/S Change in the Region	Nt Change	Region	NS/N Change in the Region	Nt Change	Region	NS/N Change in the Region
g3301a	Pol-RT	S251N	g3301a	Pol-RT	S251N	g3301a	Pol-RT	S251N
g3341a	Pol-RT	L264tta	g3341a	Pol-RT	L264tta	g3341a	Pol-RT	L264tta
g3545a	Pol-RT	Q332caa	g3545a	Pol-RT	Q332caa	g3545a	Pol-RT	Q332caa
						g4220a	Pol-RT	R557aga
g4374a	Pol-IN	A49T	g4374a	Pol-IN	A49T	g4374a	Pol-IN	A49T
c4916t	Pol-IN (SA1D2prox)	D229gat	c4916t	Pol-IN (SA1D2prox)	D229gat	c4916t	Pol-IN (SA1D2prox)	D229gat
**g5035a**	**Pol-IN** **(SA1D2prox)**	**R269Kaag**	**g5035a**	**Pol-IN** **(SA1D2prox)**	**R269Kaag**	**g5035a**	**Pol-IN** **(SA1D2prox)**	**R269Kaag**
a5702g	Vpr	E48gag	a5702g	Vpr	E48gag	a5702g	Vpr	E48gag
g6218a	Vpu	G53D						
g6280a	Vpu	A74T	g6280a	Vpu	A74T	g6280a	Vpu	A74T
	Env	M20I		Env	M20I		Env	M20I
			g6603a	Env	S128N	g6603a	Env	S128N
			7388-7402del	Env	390-394del (NSTWFdel)	7388-7402del	Env	390-394del (NSTWFdel)

Bold letters show adaptive mutations that change the Vif expression levels of a parental NL-D229gat clone. Nt, Nucleotide; NS, non-synonymous; S, synonymous.

**Table 3 viruses-13-02079-t003:** Mutations found in adapted clones with decreased Vif expression levels compared to that in the NL-R224cgc clone.

NL-cgcad-1			NL-cgcad-2			NL-cgcad-4		
Nt Change	Region	NS/S Change in the Region	Nt Change	Region	NS/N Change in the Region	Nt Change	Region	NS/N Change in the Region
						a3153t	Pol-RT	I202L
g4901c	Pol-IN (SA1D2prox)	R224cgc	g4901c	Pol-IN (SA1D2prox)	R224cgc	g4901c	Pol-IN (SA1D2prox)	R224cgc
**c4910t**	**Pol-IN** **(SA1D2prox)**	**Y227tat**	**c4910t**	**Pol-IN** **(SA1D2prox)**	**Y227tat**			
						**c4916t**	**Pol-IN** **(SA1D2prox)**	**D229gat**
						c5030t	Pol-IN (SA1D2prox)	I267att
						t5132a	Vif	I31N
t5223c	Vif	D61gac	t5223c	Vif	D61gac			
			g5310a	Vif	R90aga			
			g5593del	*1		g5593del	1	
				*2			2	
						a5613g	*2	
5632-5717del	*2							
						g5865a	Tat	K12aaa
c6633t	Env	T138I	c6633t	Env	T138I			
t6643c	Env	N141aac						
c6652t	Env	S144agt						
c6830a	Env	P204T	c6830a	Env	P204T			
g6938a	Env	V240I	g6938a	Env	V240I	g6938a	Env	V240I
			c7531t	Env	I437att			

*1, the last seven of the amino acid sequence of Vif (SHTMNGH) was changed to AIQ. *2, a large deletion in Vpr. Bold letters show adaptive mutations that change the Vif expression levels of a parental NL-R224cgc clone. Nt, Nucleotide; NS, non-synonymous; S, synonymous.

**Table 4 viruses-13-02079-t004:** Mutations found in adapted clones with decreased Vif expression levels compared to that in the NL-P238ccg clone.

NL-ccgad-1			NL-ccgad-6		
Nt Change	Region	NS/S Change in the Region	Nt Change	Region	NS/N Change in the Region
g2923a	Pol-RT	R125K			
g4071a	Pol-RT	A508T			
			g4566a	Pol-IN	V313I
			**c4916t**	**Pol-IN** **(SA1D2prox)**	**D229gat**
**a4940g**	**Pol-IN** **(SA1D2prox)**	**G237ggg**			
a4943g	Pol-IN (SA1D2prox)	P238ccg	a4943g	Pol-IN (SA1D2prox)	P238ccg
g5090a	Pol-IN	E287gaa			
	Vif	R17K			
a5242g	Vif	T68A	a5242g	Vif	T68A
a5315g	Vif	K92R			
			t5379c	Vif	D113gac
			a5602g	Vif	T188A
				Vpr	Y15C
5627-5651del	*1				
g5797a	*1				
a5820g	*1		a5820g	Vpr	R88G
g6625a	Env	K135aag	g6625a	Env	K135aag
g7032a	Env	R271K	g7032a	Env	R271K
			a7180c	Env	K320N

*1, a large deletion in Vpr. Bold letters show adaptive mutations that change the Vif expression levels of a parental NL-P238ccg clone. Nt, Nucleotide; NS, non-synonymous; S, synonymous.

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
