# Peer review of "The Expression Level of HIV-1 Vif Is Optimized by Nucleotide Changes in the Genomic SA1D2prox Region during the Viral Adaptation Process"

_viruses, 2021, doi:10.3390/v13102079_

Round 1

Reviewer 1 Report

In this manuscript, Koma and colleagues evaluate Vif adaptation mutants and assess their expression levels and impact on viral replication. The manuscript is well written and nicely describes a series of Vif mutations that affect the expression level. Some further mechanistic insight would be informative to determine whether the expression level of Vif is solely responsible for the differences in viral replication level, or if there are functional effects of the mutations. It may also be helpful in the introduction to provide more background on Vif and its roles.

Specific comments

  1. What effect do the mutations have on the function of Vif? For example, do the different Vif mutants similarly lead to degradation of APOBEC, or induce cell cycle arrest? It is unclear from the data presented whether the changes in Vif expression level are sufficient to have a functional impact during virus replication.
  2. Vif expression levels do not seem to correlate to the virus replication. For example, in Figure 3, NL-cgcad-1 appears to replicate to similar levels as WT NL4-3, yet expresses much less Vif than the WT virus. However, NL-cgcad-2, which also expresses low levels of Vif similar to the NL-cgcad-1, is profoundly impaired for viral replication. It is therefore difficult to assess whether the impact on viral infection correlates with Vif expression, or alternative factors. Addressing the comment in point 1 above may be helpful, or alternative explanations should be provided.
  3. It would be interesting to evaluate the kinetics of Vif expression. The protein expression has been measured at 24 hours post transfection, but would the Vif mutants accumulate to similar levels with time?

Minor comments

  1. In Figure 7 some graphs are missing y-axis label

Reviewer 2 Report

The manuscript entitled The Expression Level of HIV-1 Vif is Optimized by Nucleotide Changes in the Genomic SAID2prox Region during the Viral Adaptation Process submitted by Koma et al attempts to understand HIV-1 genomic changes that are critical to the expression of Vif that occur while the virus may be under selective pressure from cellularly expressed APOBEC3G (A3G).  While there are a significant amount of data presented in the submission it is not particularly clear what specific conclusion or additions to the field are added by the prolific information.   

Primary comments:

  1. The descriptive adjective “optimized” in the title implies there is a level of Vif that may be favored by the virus for robust replication. While this may be accurate, the data here do not support a discernible correlation between Vif expression and viral replication.

For example, in Figure 1, NL-gatad-5 exhibits increased Vif expression but robust viral replication is not reconstituted.  In Figure 4, NL-ccad-7 and -8 have similar Vif expression levels, but the measured growth kinetics are vastly different.  (There are a number of other examples of this disconnect between Vif expression and viral growth.)

  1. The level of expression of A3G in these H9 cells versus the level seen in patient CD4+ T cells/macrophages is not established.

  1. While the authors do demonstrate, with a significant amount of data, that the SAID2prox part of the genome is important in determining the Vif expression levels, it is not at all clear why this is an interesting finding or what the contribution to the field is in a broader context.

  1. There is an over-reliance by the authors on previous work published. For example, it would help frame the work presented here for a brief explanation on why it is hypothesized that Vif over-expression interferes with robust viral replication.

  1. While the central question of how cellular A3G expression exerts a selective pressure on viral evolution is an interesting one, the authors fall short of meaningfully addressing that query here. Perhaps a variation of A3G expression and then a comparison of resulting viral genomic mutations would more meaningfully address the question.

Secondary comments:

  1. In a number of the figures showing viral replication, there are two recurring difficulties with viewing the data: a). the wildtype virus is represented by filled-in symbols, as indicated in the legend, but in the graphs themselves there is not a matching symbol. b) Additionally, there are too many curves on the replication graphs for the information to be useful.  There must be a better way to present the work.

  1. The method used to quantitate RT activity is not clearly delineated anywhere in the paper. It should probably be articulated in the Materials and Methods section.

  1. The phrase “growth potential” used by the authors throughout the manuscript is ambiguous. Perhaps “delayed kinetics” is a more accurate phrasing?

  1. In Figure 7, the switching of colors to represent various clones is confusing. To aid the reader, the color of wildtype virus, at the very least, should be the same color in all the graphs (B & C). 

  1. It is not entirely clear why the authors have selected to examine protease-inhibitor resistant and integrase-inhibitor resistant patients to examine. Why is it anticipated that the emergence of either of these types of mutations would be related to their central question of interest of A3G-driven mutations that impact Vif expression?

  1. Only one lab has suggested a Vif-deficient virus can adapt to grow in the presence of A3G expression. This has not been repeated and is controversial in the field.  In addition Vif-deficient sequences recovered from patients is incredibly rare.  Arguments of relevance based on this somewhat weak foundation decreases reader confidence in the relevance of the data.

Round 2

Reviewer 1 Report

The authors have nicely addressed the reviewer concerns, and the manuscript is well-written. No further revisions are necessary.